# Contribution of Dynamic and Genetic Tests for Short Stature Diagnosing: A Case Report

**DOI:** 10.3390/diagnostics13132259

**Published:** 2023-07-04

**Authors:** Betina Biagetti, Irene Valenzuela, Ariadna Campos-Martorell, Berta Campos, Sara Hernandez, Marina Giralt, Noelia Díaz-Troyano, Emilio Iniesta-Serrano, Diego Yeste, Rafael Simó

**Affiliations:** 1Endocrinology Department, Diabetes and Metabolism Research Unit, Vall d’Hebron University Hospital and Vall d’Hebron Research Institute (VHIR), Universidad Autónoma de Barcelona, Reference Networks (ERN), 08035 Barcelona, Spain; 2Department of Clinical and Molecular Genetics and Rare Disease Unit and Medicine Genetics Group, Vall Hebron Research Institute, 08035 Barcelona, Spain; 3Pediatric Endocrinology Section, Vall d’Hebron University Hospital and Vall d’Hebron Research Institute (VHIR), Universidad Autónoma de Barcelona, 08193 Barcelona, Spaindiego.yeste@vallhebron.cat (D.Y.); 4Department of Biochemistry, Vall d’Hebron University Hospital, 08035 Barcelona, Spain; 5Pharmacy Department, Vall d’Hebron University Hospital, 08035 Barcelona, Spain; 6CIBER Enfermedades Raras, Instituto Carlos III, 28220 Madrid, Spain

**Keywords:** short stature, genetics, macimorelin, *GHS-R*, *ADAMTS*, *ADAMTS17*

## Abstract

Genetic tests have led to the discovery of many novel genetic variants related to growth failure, but the clinical significance of some results is not always easy to establish. The aim of this report is to describe both clinical phenotype and genetic characteristics in an adult patient with short stature associated with a homozygous variant in disintegrin and metalloproteinase with thrombospondin motifs type 17 gene (*ADAMTS17*) combined with a homozygous variant in the GH secretagogue receptor (*GHS-R*). The index case had severe short stature (SS) (−3.0 SD), small hands and feet, associated with eye disturbances. Genetic tests revealed homozygous compounds for *ADAMTS17* responsible for Weill–Marchesani-like syndrome but a homozygous variant in *GHS-R* was also detected. Dynamic stimulation with an insulin tolerance test showed a normal elevation of GH, while the GH response to macimorelin stimulus was totally flattened. We show the implication of the *GHS-R* variant and review the molecular mechanisms of both entities. These results allowed us to better interpret the phenotypic spectrum, associated co-morbidities, its implications in dynamic tests, genetic counselling and treatment options not only to the index case but also for her relatives.

## 1. Introduction

Short stature (SS) is a relative common consultation in the pediatric setting. It is defined as a height lower than the 3rd percentile for the mean height of a given age, sex, and population group. A large number of children with SS after careful evaluation remain without a definitive diagnosis and are labelled as having idiopathic short stature (ISS) [1,2].

In the adult setting SS concern presents less frequently. Although the epiphyseal growth plate closely determines that growth hormone (GH) treatment will be ineffective in improving height in adults, a familial history of SS should be studied looking for genetic defects because of the potential consequences on younger relatives and offspring.

In the last few years, there has been a change regarding our understanding of the growth process from a traditional view centered on the GH-IGF-1 axis to a growth plate-centered view as the structure responsible for height gain [3]. Normal extrauterine growth requires normal concentrations of GH and IGF-I but also the normal production and action of multiple other hormones, paracrine factors, and extracellular matrix molecules for chondrocyte proliferation, hypertrophy, and extracellular matrix production [3]. 

We report a clinical case of a 19-year-old female with SS, in which the genetic test revealed two homozygous variants, one in disintegrin and metalloproteinase with thrombospondin motifs type 17 gene (*ADAMTS17*) and the other in the GH secretagogue receptor (*GHS-R*), both related to growth. We explain how dynamic tests allowed us to better interpret the phenotypic spectrum in the index case, the repercussions for her siblings and the implications on dynamic tests.

### Case Presentation

A 19-year-old female, 132 cm (<3SD) in height, was referred to the adult endocrinology clinicians for SS. She was originally from Bolivia and was the older child of four siblings (she was unaware of the paternal history and did not have contact with him). She was born at full term and her birth length and weight were average. Her past history was unremarkable except for high myopia. The menarche was at 12 years old, and she had regular cycles. X-rays, as expected, showed a closed epiphyseal, but considering she had a family history of SS in several family members including her mother, we decided to perform hormonal and genetic studies. The family pedigree and genetic results are shown in Figure 1.

## 2. Materials and Methods

### 2.1. Ethical Statement

Written informed consent was obtained from the index case and her mother, as legal guardian of the siblings, for genetic tests, dynamic studies and publication. We followed standard clinical practice without any diagnostic tests or treatment out of our usual protocols. Therefore, following the regulations of our hospital the institutional review board, approval was not required.

### 2.2. Biochemical Assays

IGF-1 concentrations were measured using a Liaison XL IGF-1 chemiluminescence assay (Liaison XL-Diasorin, DiaSorin, Saluggia, Italy). The assay is referenced against the first World Health Organization (WHO) International Standard for IGF-1. According to the National Institute for Biological Standards and Control (NIBSC) code 02/254, the limit of detection was 3 ng/mL. GH was measured by Cobas e 801 (Roche Diagnostics, Muttenz, Switzerland); the assay is referenced against the second WHO NIBSC International Standard 98/574 with an analytical sensitivity of 0.03 ng/mL. Glucose and cortisol were measured by chemiluminescence (Atellica Solutions, Siemens Healthcare Diagnostics, EUA).

### 2.3. Genetic Tests

Analysis of the clinical examination of the proband did not suggest any specific diagnosis. Microarrays did not detect causative copy number variants. To uncover the underlying molecular cause, we performed whole exome sequencing (WES) in DNA extracted from peripheral blood. Exome capture was performed using the Nimblegen SeqCap EZ MedExome capture kit (Roche Diagnostics) and the library was sequenced on an Illumina Hiseq 2000 platform. Paired-end sequences were obtained with a read length of 200 bp and a mean coverage depth of 90×. An in-house bioinformatics pipeline based on the GenomeAnalysisToolkit best practice guidelines was used for the identification of single nucleotide variants and small insertions and deletions, after comparison with the reference human genome (GRCh37). Finally, we performed RNA analysis via quantitative reverse transcription PCR (RT-qPCR) in order to assess the effect of the *ADAMTS17* intronic variant in the splicing process.

### 2.4. Dynamic Tests

The insulin tolerance test (ITT) stimulates GH secretion through growth hormone-releasing hormone receptor (GHRH-R) activation and is considered the gold standard for GH deficiency (GHD) evaluation [4,5,6]. Insulin was given at 0.15 unit/kg subcutaneously and blood samples were drawn for serum blood glucose, cortisol and serum growth hormone at 0, 30, 60, 90, and 120 min after giving insulin. We also checked all blood glucose levels on a glucometer during ITT for hypoglycemia. At these time intervals, a blood sugar level < 50 mg/dL was deemed hypoglycemic. The cut-off level of GH < 10 ng/mL is considered the gold standard for growth hormone deficiency.

Macimorelin is an orally active ghrelin mimetic that binds to the GHS-R with a similar affinity to ghrelin. It was approved in the USA and some European countries in 2019 for use as a diagnostic test for GHD in adults. Its effectiveness was compared to the GHRH plus arginine test [7] and ITT [8]. The cut-off of 2.8 μg/L at 45 min is considered for adults [8] and 7 ng/mL for pediatric patients [9]. After an overnight fast, macimorelin (0.5 mg/kg) was given. Blood samples for GH levels were taken at 0, 30, 60 and 90 min after medication.

## 3. Results

### 3.1. Baseline Phenotype

As explained above, the index case was a 19-year-old female, height 132 cm < 3 SD and weight 42.0 Kg, and thus categorized as extremely SS (i.e., small hands and feet, harmonic SS). The mid-parental height calculated by the Tanner method (average of the father and mother’s height minus 6.5 cm) was 139.5 cm. She did not present with evidence of systemic, endocrine or nutritional abnormalities. She had eye disturbances comprising high myopia, and ophthalmologic examination revealed a shallow anterior chamber, lens subluxation, spherophakia, and fundus albipunctatus in both eyes. She did not have joint stiffness or cardiac alterations. 

She had a baseline GH of 0.42 ng/mL (range: 0–9.9 ng/mL), and an IGF-1 of 135 ng/mL (range: 191–483 ng/mL), (−2.80 SDS). A second determination confirmed normal GH 1.2 ng/mL with low IGF-1 of 145 ng/mL (−2.75 SDS). 

The 11-year-old half-brother, who had both variants in heterozygosis, had a height of 132.6 cm (p2, −2.21 SD) and a weight of 48.5 Kg (p73, 0.62 SD), a baseline GH of 0.12 ng/mL (range: 0–6.3), and a IGF-1 of 66 ng/mL (range: 37–459) (−1.38 SDS).

The 9-year-old sister who only had a heterozygous variant in GHS-R, had a height of 126.6 cm (p3, −1.93 SD) and a weight of 34.15 Kg (p43, −0.18 SD). Her baseline GH was 0.74 ng/mL (range: 0–7.8), and a IGF of 1.74 ng/mL (range: 49–451) (SDS) (−1.47 SDS).

### 3.2. Genetics Tests

The family’s genetic results are shown in Figure 1. By WES analysis we identified a homozygous missense variant of unknown clinical significance in the *GHS-R* gene, NM_198407.2:c.269T > C, p.(Leu90Pro). In the *ADAMTS17* gene we found two homozygous rare variants: NM_139057.4:c.1323-11G > A and NM_139057.4:c.2485C > T, p.(Arg829Trp). Nevertheless, the latter is considered to not have clinical relevance. Co-segregation studies showed that the *GHS-R* variant was present in a heterozygotic state in the 10-year-old half-brother, the 9-year-old half-sister and the mother of the proband in Figure 1. The *ADAMTS17* intronic variant was present in a heterozygotic state in the mother and half-brother. The father’s sample (consanguineous) was not available.

We did not find any other variants that could be linked with the phenotype of the index case.

The *GHS-R* p.(Leu90Pro) variant was not found in the gnomAD database (version 2.1.1) it had a REVEL score of 0.62, not predicting a damaging impact on protein function. On the other hand, the *ADAMTS17* c.1323-11G > A variant was found in a heterozygotic state in 7/141,405 individuals of the gnomAD database. GnomAD’s inferred phasing data showed that c.1323-11G > A and p.(Arg829Trp) variants are on the same copy of the gene (*cis*) in all seven individuals. The c.1323-11G > A intronic variant had a high probability of being spliceogenic according to the splicing prediction tool SpliceAI (score = 0.85). Subsequent functional RNA studies showed that this variant produced two alternative transcripts, one leading to a loss-of-function protein, p.(Lys441AsnfsTer3), and the other to the deletion of lysine 441 and the insertion of four new amino acids in frame, p.(Lys441delinsAsnCysPheArg). On the contrary, in silico tools did not predict a deleterious effect of the missense variant p.(Arg829Trp) in the protein (REVEL score = 0.3), and it is thus considered a benign variant.

### 3.3. Dynamic Tests 

GHRH-R provocative tests, ITTs, in the adult index and GHS-R provocative tests with macimorelin in the index case and siblings were performed to evaluate the hypothesis that GH secretion by the GHRH-R pathway (ITT) would be unperturbed, but the GH secretion by GHS-R stimulated by macimorelin would be affected. 

The provocative ITT in the index case showed a GH peak of 21.7 ng/mL (normal values > 10 ng/mL) after a hypoglycemia of 34.0 mg/dL.

With the macimorelin test, a flat GH response was observed in the index case which had a homozygotic variant of GHS-R (baseline GH of 0.14 ng/mL and at 90 min of 0.06 ng/mL) and a blunted peak of GH in the siblings (which had a GHS-R variant in heterozygosis). GH increased up to 4.08 ng/mL in the brother and up to 6.23 ng/mL in the sister (Figure 2).

## 4. Discussion

In this clinical case, we describe the phenotype associated with one variant related with growth, *ADAMTS17,* which belongs to a group of proteins located in the cellular matrix related to cell growth and survival and linked with SS in the context of both Weill–Marchesani syndrome (WMS) [10] and Weill–Marchesani-like syndrome (WMLS) [11]. Additionally, a homozygotic variant, of unknown clinical significance in *GHS-R*, responsible for improving GH pulsatility and amplitude [12], was also detected. The dynamic stimulation of GHS-R allowed us to interpret the GHS-R co-implication in the SS phenotype of the index patient and her pre-puberal siblings.

Human height has a high degree of heritability, indicating that genetic factors are one of the main determinants [13]. External conditions also impact height variability such as fetal growth restriction [14] and nutrition [15]. Studies performed in twins showed an increasing pattern of height heritability with age [13]. GWA studies have identified more than 12,000 common variants affecting adult height [16]. Extremes in height are often caused by monogenic mutations in one of the genes critical for growth control. In our clinical case *ADAMTS* is the critical one. However, height is a classic polygenic trait and a small list of loci (including GHS-R) with significant *p* values related to the final height have been identified as being involved in skeletal and/or growth syndrome [17]. To achieve normal growth, besides the growth plate`s matrix protein, normal concentrations of GH and IGF-I are required [3]. 

As mentioned above, the protein ADAMTS participates in tissue morphogenesis. Mutations in certain family members result in inherited genetic disorders, such as WMS, while the aberrant expression or function of others is associated with arthritis, cancer and cardiovascular disease [18]. WMS is a genetically heterogeneous disorder affecting connective tissue, the growth plate, etc., consequently impacting stature. Mutations in *ADAMTS10*, *ADAMTS17*, *LTP2* cause the autosomal recessive and mutations in the *FBN1*, the autosomal dominant form of the disorder [19]. In particular, *ADAMTS17* variants have been connected with several incomplete forms of WMS called WMLS [10,11,20,21], a rare connective-tissue disorder characterized by SS, eye disturbances consisting of lens subluxation, spherophakia, severe myopia and possible glaucoma secondary to shallow anterior chamber angles, but lacking joint stiffness, brachydactyly, and cardiac valvular abnormalities [10] as in our patient.

Regarding the GH–IGF1 system, GH effects on long bone growth are independent of IGF-1 levels. GH absence seems to affect growth more deeply (since both GH and GH-stimulated IGF-I effects are absent) than IGF-1. For example, tibial linear growth rate was reduced by approximately 35% in IGF-1-null mice and by about 65% in GH-R-null mice between post-natal days 20 and 40, a time where peak GH affects normal longitudinal growth in mice [22].

Intriguingly, there is an ancestral link between food and GH (stomach and growth). Fasting activates GH, while feeding inhibits it. Moreover, GH secretion is regulated directly by changing some nutrient and food-related factors (Figure 3). The hypothalamic and peripheral somatostatin secreted by delta cells of the pancreas, hyperglycemia [23], free fatty acids [24] and IGF-1 [25] inhibit GH and GHRH secretion. On the other hand, ghrelin, secreted in the stomach [26], hypoglycemia [27] and amino acids [28] are physiological stimuli of GH.

Figure 4 schematically represents the somatotroph cell. There are two predominant receptors that stimulate GH secretion, the GH-releasing hormone receptor and the GH secretagogue receptor, or ghrelin receptor. Both are G protein-coupled receptors and stimulate growth hormone secretion under nutritional or physiological challenges. Particularly, ghrelin is a brain–gut peptide that acts through GHS-R [26]. Beyond its action in energy metabolism and regulation of appetite, this peptide is related to bone formation and growth hormone release [29,30]. Ghrelin acts on both the somatotrophs of the anterior pituitary gland and GHRH-secreting neurons and on somatostatin-secreting neurons (opposing the action of somatostatin) in the hypothalamus via GHS-R. GH released after the administration of ghrelin is much greater than that induced by maximal dosages of GHRH [31]. Ghrelin and GHRH have a synergistic effect on the secretion of GH when they are administered simultaneously [31]. Thus, ghrelin is a positive modulator of GH secretion.

Pantel et al. [32] first demonstrated the functionally significant GHS-R mutation in two unrelated families from Morocco. They studied GHS-R variants in two groups of patients presenting with isolated GHD (n = 51) or with ISS (n = 41). They found a missense variant (p.A240E) which resulted in decreased cell surface expression of the receptor. The patients presented with short stature, with or without isolated GH deficiency, and some were overweight or obese. In vitro functional studies of this mutation showed selective impairment of its constitutive activity but preserved the ability to respond to ghrelin. The same group [33] also reported a patient with partial, isolated GHD, carrying two new GHS-R genetic defects, born from two heterozygotic parents of normal stature. In vitro experiments resulted in a partial loss of constitutive activity of the receptor, whereas both its ability to respond to ghrelin and its cell surface expression were preserved. 

Similarly, GHSR-null mice have lower IGF-I levels when compared to wild-type animals, while ghrelin-null mice have a tendency to lower IGF-I concentrations when compared to control animals [12]. Moreover, the application of a GHSR antagonist results in a decrease in GH pulse amplitude in rodents [34].

As mentioned above, in the general population macimorelin (a ghrelin mimetic) is used for adult GHD diagnosis [8]. Furthermore, studies with macimorelin in the pediatric population with suspected GHD [9] show comparable accuracy to insulin tolerance tests. Thus, macimorelin binds to GHS-R and stimulates dose-dependent increases in GH levels, used to assess GH production in patients with a suspected GH deficiency. This response was affected in this family report. In our index patient we found by WES a homozygotic missense variant of unknown clinical significance in GHS-R NM_198407.2:c.269T > C, p.(Leu90Pro). The baseline IGF-1 levels were also systematically lower (−2.5 SDS). ITT was normal but the dynamic stimulation of the GHS-R with macimorelin showed a flat GH response. The siblings with the GHS-R heterozygotic variant had normal baseline IGF-1 levels but an impaired macimorelin response. The clinical significance of the homozygotic variant found in GHS-R was evaluated using a dynamic test with macimorelin. Macimorelin administration showed a flat GH response in the index case (homozygous) and a blunted peak of GH in the siblings (heterozygous), thus suggesting a GHSR variant may be responsible for the abnormal response. However, further studies are needed to establish its functional implications and impact on growth.

Recognizing whether a GHS-R variant is clinically relevant is crucial because patients with homozygotic variants could have impaired GH secretion (ghrelin is a positive modulator of GH secretion); however, standard dynamic tests (ITT or DOPA) work by GHRH-R pathways and will be unaffected. Thus, these patients could be misdiagnosed and not benefit from GH treatment based on classic dynamic tests.

Therefore, our patient has two defects that impact on their stature, the alteration of connective tissue (e.g., growth plate) from the *ADMTS17* variant and impaired GH secretion from the GHSR variant. Both variants could contribute to the extreme SS phenotype observed in the index case. However, more comprehensive investigations are needed to establish a clear genotype–phenotype correlation.

The diagnosis of both variants was helpful. For the patient it offered her a better follow-up related to *ADAMTS17*, such as supportive treatment for eye disturbances and other co-morbidities. For her siblings the genetic study ruled out WMLS (as they were heterozygous) and uncovered a GHS-R variant which is important in order to initiate GH treatment if required and avoid the misdiagnosis of GHD.

Finally, two main learning points can be taken from the present study: (1) Because of the potential consequences on younger relatives and offspring it is relevant to perform genetic testing in adult patients with SS and familial history of SS and/or syndromic features and; (2) identify GHS-R mutations with clinical relevance, because patients with this mutation exhibit an unaffected response to standard dynamics tests (e.g., insulin tolerance test) which work by the GHRH-R pathways. Thus, a patient with a GHS-R variant could be misdiagnosed and would not obtain the benefits from GH treatment if required.

Our results have to be taken with caution, a hormonal dynamic test is not a standard approach for assessing the functional significance of genetic variants and we lack a control group. Moreover, due to the extreme rarity of the co-existence in a single patient of these two rare variants, larger samples to evaluate our results will be extremely difficult to find. Therefore, further research is needed to establish concrete recommendations for patient management based on these genetic findings.

## 5. Conclusions

To the best of our knowledge, this is the first report of a patient with two extremely rare variants in a homozygous state, *ADAMTS17* and *GHS-R*. The identification of the *ADAMTS17* mutation confirmed the diagnosis of WMLS, a rare genetic disorder characterized by SS, brachydactyly and characteristic eye abnormalities present in our patient. The clinical significance of the homozygotic variant found in *GHS-R* was evaluated using macimorelin (a ghrelin analogue), showing a flat GH response in the index case (homozygous for GHSR) and a blunted peak of GH in the siblings (heterozygous).Therefore, our patient has two defects that impact stature, the WMLS which affects the connective tissue (e.g., the growth plate) from the homozygous *ADMTS17* variant and the impaired GH secretion from a homozygous GHSR variant. Both variants could contribute to the extreme SS phenotype observed. The genetic diagnosis in this adult patient was relevant to a better follow-up of her and her siblings, due to the implications of the *GHS-R* mutation in the stimulation tests which not only have clinical implications but also led to genetic counselling.

## Figures and Tables

**Figure 1 diagnostics-13-02259-f001:**
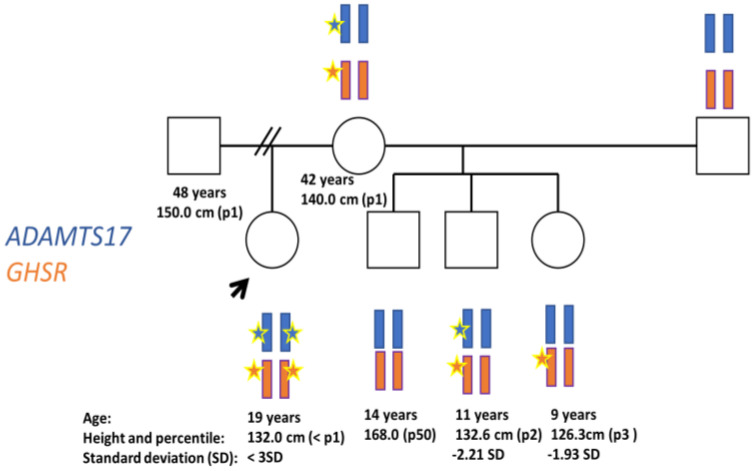
Pedigree illustrating *ADAMTS* and *GHS-R* inheritance. Circles and squares denote female and male family members, respectively. The index case is indicated by arrows. Stars denote mutation.

**Figure 2 diagnostics-13-02259-f002:**
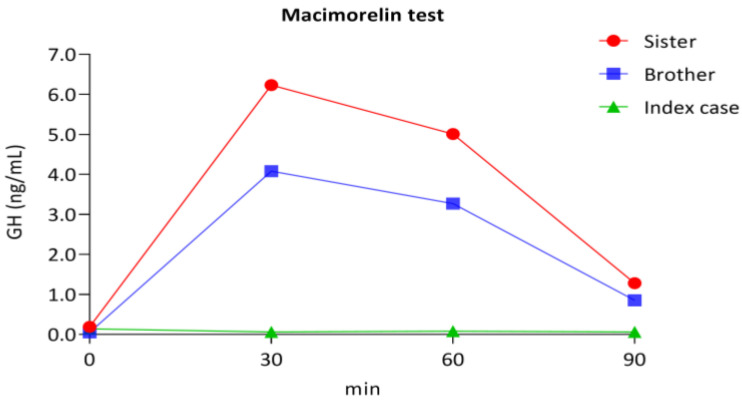
Response to macimorelin. The figure shows a timeline with GH levels after macimorelin administration. The index case with the homozygous mutation in GHS-R had a flattened response to macimiorelin and both siblings (heterozygous) with the GHS-R had an impaired response.

**Figure 3 diagnostics-13-02259-f003:**
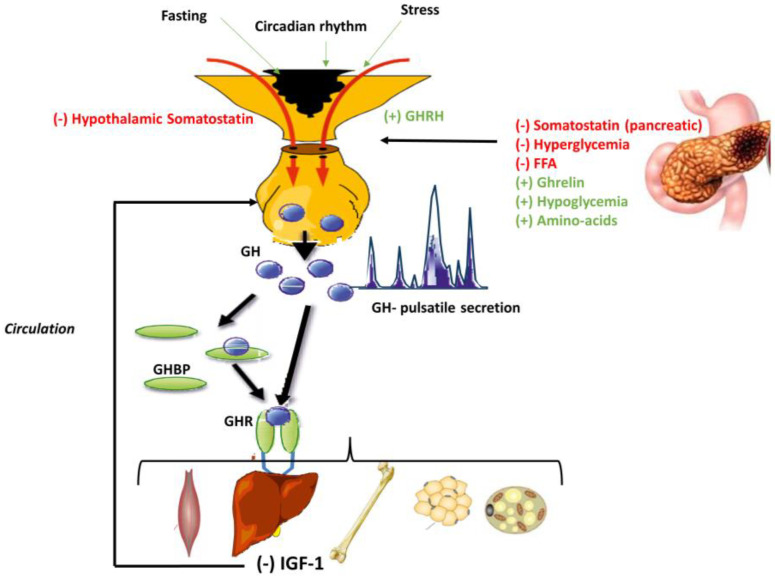
GH secretion and the link between food and GH. GH synthesis is controlled by hypothalamic factors, including the growth hormone-releasing hormone (GHRH) which stimulates GH secretion, and somatostatin which is the main inhibitory stimulus for GH release. At the extra-hypothalamic level, somatostatin, secreted by delta cells of the pancreas, along with hyperglycemia, free fatty acids (FFA) and the negative feedback of IGF-1, inhibit GH secretion. On the contrary, ghrelin, secreted in the stomach, hypoglycemia and amino acids, stimulate GH secretion.

**Figure 4 diagnostics-13-02259-f004:**
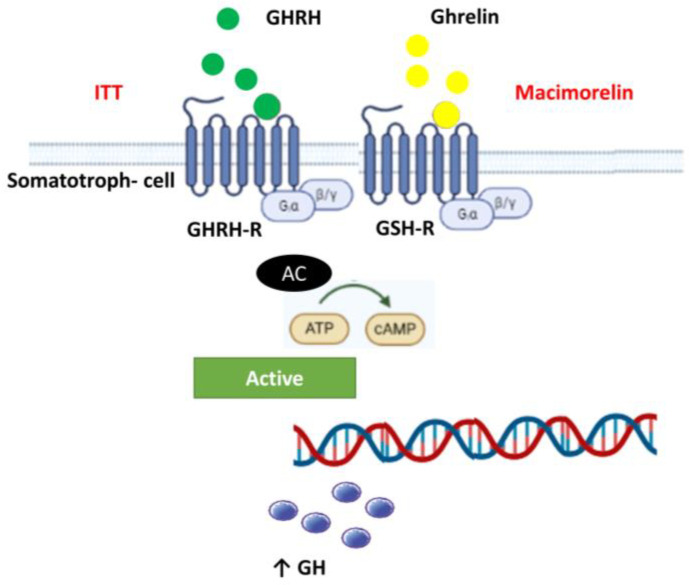
Somatotroph cell and GHRH and ghrelin receptors. Schematic representation of the somatotroph cell in the pituitary gland. There are two predominant receptors that stimulate GH secretion, the GH-releasing hormone receptor (GHRH-R) and the GH secretagogue receptor (GHS-R or ghrelin receptor). Both of them are G protein-coupled receptors. Macimorelin binds to GHS-R and amplifies GH secretion. Hypoglycemia after ITT: (insulin tolerance test) stimulates GHRH, which is the main stimuli for GH secretion.

## Data Availability

Further inquiries can be directed to the corresponding author.

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
