# Peer review of "Contribution of Dynamic and Genetic Tests for Short Stature Diagnosing: A Case Report"

_diagnostics, 2023, doi:10.3390/diagnostics13132259_

Round 1

Reviewer 1 Report

Introductory paragraph: The authors present an interesting study on a 19-year-old female patient with short stature and other congenital malformations. The authors carried out clinical laboratory experiments and whole exome sequencing studies. The authors observed two sets of variants in ADAMTS17 and another variant in GHS-R. However, the attempt by the authors to mechanistically link variants in GHS-R and observations in the hormonal experiments is premature and problematic, as these observations may just be co-occurring but not necessarily related. Again, this approach is not the standard for validating the pathogenicity or otherwise of genetic variants.

 General comments:

Major points:

·       Page 2 lines 70 to 72: The authors are silent on Institutional Review Board (IRB) approval for the study, though they present information on consent. The authors may want to add a statement stating that IRB approval was obtained (if indeed that is the case) and include the approval number and the IRB that gave the approval. It will not be acceptable to carry out such human subject research without IRB approval.

·       Apart from rare variants observed in ADAMTS17 and GHS-R, the authors may want to present other rare variants that were observed as supplementary material. The authors seem to be preoccupied with variants in ADAMTS17 and GHS-R genes because variants in these genes have previously been associated with the phenotypes described in the proband in the current study. This approach, however, limits discovery.

·       Page 7 lines 269 to 271: the following conclusion by the authors is premature and problematic: “The clinical significance of the homozygous variant found in GHS-R was established using macimorelin (ghrelin analogue) which showed a flat GH response in the index case (homozygous) and a blunted peak of GH in the siblings (heterozygous)”. This is not an acceptable approach for delineating the functional significance of genetic variants! For example, functional experiments in zebrafish or cell migration assays using the mutant mRNA may help us ascertain the functional effect of the mutant mRNA. Using clinical laboratory observations to deduce the effect of genetic variants is strange! How do you prove that the clinical observations are due to the genetic variants in the GHS-R gene but no other gene variants? Alternatively, the authors may want to go through literature and databases such as Mouse Genome Informatics (MGI) to ascertain the functional effects of such variants.

·       Page 7 lines 272 to 273: the following conclusion by the authors lacks merit: “Therefore, both genes contribute to the extreme SS phenotype observed 272 in the index case”. The author correctly states in the result section that the variant in GHS-R had a REVEL score of 0.62, indicative of a benign variant. So, I wonder why the authors are now postulating that variants in both genes contribute to the phenotype! How do you establish the contribution of both genes to the phenotype? Importantly, the authors state that “The identification of ADAMTS17 mutation confirmed the diagnosis of Weill-Marchesani-like syndrome, a rare genetic disorder characterized by short stature, brachydactyly and characteristic eye abnormalities present in our patient”. Thus, what is the probable contribution of the GHS-R variant to the Weill-Marchesani-like syndrome?

 Minor points:

·       Functional test: in terms of genetics and genomics, the hormonal experiments conducted by the authors do not constitute functional experiments. Since the authors carried out no study in animal or cell models, or even gene expression experiments, it is erroneous to use the term “functional” in the title, result, discussion, and elsewhere in the manuscript in relation to the effect of the GHS-R and other variants observed.

·       Page 7 lines 274 to 275: this statement is problematic: “due to the implications of GHS-R mutation in the stimulation tests”. How can a benign variant be responsible for these clinical observations?

·       The authors may want to use the term “variants” instead of “mutation” in the entire manuscript. The current consensus is that mutation should be used to describe the process that generates the genetic variants.

 Specific comments:

Page 1 line 18: remove the comma after “both”.

 Page 2 line 58: replace “high” with “height”.

 Page 2 line 65: replace “high” in the figure legend with “height”.

 Page 3 line 110:  Insert “it” before the “was”.

 Page 3 line 122: delete the comma after “nutritional”.

 Page 3 line 123: delete the comma after “revealed”.

Page 3 line 125: you may want to insert a semi-colon after the “SDS”.

 Page 3 lines 126 to 134: punctuations in these lines are poorly done.

 Page 5 line 212: “Figure 3. Material)” may be presented as (Figure 3).

 Page 8 line 303: the authors state that “Institutional Review Board Statement: Not applicable”. I sis the case that this research involving human subjects was carried out without IRB approval?

 Page 8 line 306: delete “Conflict of Interest”.

 Page 8 lines 322 to 325: change the capital letters to sentence cases to reflect the formatting style for the references.

Author Response

#Reviewer 1#

Introductory paragraph: The authors present an interesting study on a 19-year-old female patient with short stature and other congenital malformations. The authors carried out clinical laboratory experiments and whole exome sequencing studies. The authors observed two sets of variants in ADAMTS17 and another variant in GHS-R. However, the attempt by the authors to mechanistically link variants in GHS-R and observations in the hormonal experiments is premature and problematic, as these observations may just be co-occurring but not necessarily related. Again, this approach is not the standard for validating the pathogenicity or otherwise of genetic variants.

ANSWER: Many thanks for the revision process of our manuscript and all your suggestions. We feel that your feedback has strengthened our paper.

We understand the referee’s concerns about our statements regarding the link between the GHSR mutation and the hormonal response. We agree that the hormonal dynamic test is not a standard approach for assessing the functional significance of genetic variants.

However,  Ghrelin is the endogenous ligand for the GHS-R [also called the ghrelin receptor] (1,2). Pharmacological treatment of rats and mice with ghrelin increases GH secretion, and GHSR-/- mice are refractory to the stimulatory effects of ghrelin on GH release, confirming that the GHS-R is the physiologically relevant ghrelin receptor mediating GH secretion (3).

Synthetic agonists of this receptor, known as ghrelin mimetics or GH secretagogues, are molecules that evoke dose-dependent increases in GH levels. Macimorelin binds the GHSR receptor to pituitary and hypothalamic extracts with an affinity similar to ghrelin(4,5). Therefore, the flat GH response in the index case (homozygous for GHSR) and the blunted peak of GH in the siblings (heterozygous) argue in favor of  this variant inhibiting the GHSR stimulatory effects of macimorelin on GH release throughout GHSR.  We have added a summary of results in mice on page 7 lines 263-266 and we have noted the Reviewer’s concern regarding dynamic tests in page 7 lines 278-283, and page 8  lines 309-314.

  1. Kojima M, Hosoda H, Date Y, Nakazato M, Matsuo H, Kangawa K. Ghrelin is a growth-hormone-releasing acylated peptide from stomach. Nature 1999;402(6762):656–660.
  2. Piccoli F, Degen L, MacLean C, Peter S, Baselgia L, Larsen F, Beglinger C, Drewe J. Pharmacokinetics and pharmacodynamic effects of an oral ghrelin agonist in healthy subjects. J Clin Endocrinol Metab 2007;92(5):1814–1820.
  3. Sun Y, Wang P, Zheng H, Smith RG. Ghrelin stimulation of growth hormone release and appetite is mediated through the growth hormone secretagogue receptor. Proc Natl Acad Sci U S A 2004;101(13):4679–4684.
  4. Broglio F, Boutignon F, Benso A, Gottero C, Prodam F, Arvat E, Ghè C, Catapano F, Torsello A, Locatelli V, Muccioli G, Boeglin D, Guerlavais V, Fehrentz JA, Martinez J, Ghigo E, Deghenghi R. EP1572: a novel peptido-mimetic GH secretagogue with potent and selective GH-releasing activity in man. J Endocrinol Invest 2002;25(8):RC26-28.
  5. Garcia JM, Swerdloff R, Wang C, Kyle M, Kipnes M, Biller BMK, Cook D, Yuen KCJ, Bonert V, Dobs A, Molitch ME, Merriam GR. Macimorelin (AEZS-130)-stimulated growth hormone (GH) test: validation of a novel oral stimulation test for the diagnosis of adult GH deficiency. J Clin Endocrinol Metab 2013;98(6):2422–2429.

 General comments:

Major points:

  • Page 2 lines 70 to 72: The authors are silent on Institutional Review Board (IRB) approval for the study, though they present information on consent. The authors may want to add a statement stating that IRB approval was obtained (if indeed that is the case) and include the approval number and the IRB that gave the approval. It will not be acceptable to carry out such human subject research without IRB approval.

ANSWER: In accomplishment of the regulations of our Hospital, this type of study only requires approval by the patient and or legals guardians. We have added the following sentence to the revised manuscript: The study was conducted according to the mandates of the Declaration of Helsinki and good clinical practices. The patients’ confidential information was protected according to the Spanish national data protection law”.

  • Apart from rare variants observed in ADAMTS17and GHS-R, the authors may want to present other rare variants that were observed as supplementary material. The authors seem to be preoccupied with variants in ADAMTS17 and GHS-R genes because variants in these genes have previously been associated with the phenotypes described in the proband in the current study. This approach, however, limits discovery.

ANSWER: In this regard, first we performed a targeted approach for some candidates genes related with hight  ACAN, BRAF, BTK, CBL, CCDC8, COL10A1, COL11A1, COL11A2, COL1A1, COL2A1, COL9A1, COL9A2, COL9A3, CUL7, EVC, FBN1, FGFR3, GH1, GHR, GHSR, GHRHR, GLI2, GPC3, H19, HESX1, HRAS, IGF1, IGF2, IGF1R, IGFALS, IHH, KRAS, LHX3, LHX4, MAP2K1, MAP2K2, MRAS, NF1, NPPC, NRAS, OBSL1, OTX2, POU1F1, PPP1CB, PROKR2, PROP1, PTPN11, RAF1, RIT1, RRAS, SHOC2, SHOX, SOS1, SOS2, SOX3, SPRED1, SRCAP, STAT5B -> here we identify the GHSR variant. Then we followed with the Human Phenotype Ontology terms HP:0004322, HP:0032367, HP:0011003, HP:0004209, HP:0001156 -> here we identify the ADMATS17 variant. Finally in the exome extension (non-target study) we did not identify any other variant than ADAMTS17 or GHS-R that could be linked with the phenotype. We have amended the text page 4 line 151-152

  • Page 7 lines 269 to 271: the following conclusion by the authors is premature and problematic: “The clinical significance of the homozygous variant found in GHS-Rwas established using macimorelin (ghrelin analogue) which showed a flat GH response in the index case (homozygous) and a blunted peak of GH in the siblings (heterozygous)”. This is not an acceptable approach for delineating the functional significance of genetic variants! For example, functional experiments in zebrafish or cell migration assays using the mutant mRNA may help us ascertain the functional effect of the mutant mRNA. Using clinical laboratory observations to deduce the effect of genetic variants is strange! How do you prove that the clinical observations are due to the genetic variants in the GHS-R gene but no other gene variants? Alternatively, the authors may want to go through literature and databases such as Mouse Genome Informatics (MGI) to ascertain the functional effects of such variants.

ANSWER: As we explained above, we do not find any other variant in the non-target study of the exome. Taking into account that macimorelin is a ghrelin mimetic which binds the GHSR receptor in the pituitary and hypothalam with an affinity similar to ghrelin, the flat GH response obtained in the index case (homozygous for GHSR) and a blunted peak of GH in the siblings (heterozygous) points to this GHSR variant as the cause of these results. Nevertheless, we have tone done and re-wrote this paragraph as follows: “ The clinical significance of the homozygous variant found in GHS-R was evaluated using dynamic test with macimorelin. This is a ghrelin analogue which binds the GHSR receptor on the pituitary and hypothalamus with an affinity similar to ghrelin. Macimorelin administration showed a flat GH response in the index case (homozygous) and a blunted peak of GH in the siblings (heterozygous), thus suggesting GHSR variant as the responsible of this abnormal response. However, further studies are needed to establish its functional implications and impact on growth. Page 7 and 8 lines 278-283 of the revised manuscript

  • Page 7 lines 272 to 273: the following conclusion by the authors lacks merit: “Therefore, both genes contribute to the extreme SS phenotype observed in the index case”. The author correctly states in the result section that the variant in GHS-R had a REVEL score of 0.62, indicative of a benign variant. So, I wonder why the authors are now postulating that variants in both genes contribute to the phenotype! How do you establish the contribution of both genes to the phenotype? Importantly, the authors state that “The identification of ADAMTS17 mutation confirmed the diagnosis of Weill-Marchesani-like syndrome, a rare genetic disorder characterized by short stature, brachydactyly and characteristic eye abnormalities present in our patient”. Thus, what is the probable contribution of the GHS-R variant to the Weill-Marchesani-like syndrome?

ANSWER: We agree with the referee that the conclusion merit a better explanation.  We have amended as follow “Therefore, our patient has two defects that impact on stature, the alteration of the connective tissue (i.g. growth plate) from ADMTS17 variant and the impaired GH se-cretion from GHSR variant, both variants could contribute to the extreme SS phenotype observed in the index case. However, more comprehensive investigations are necessary to establish a clear genotype-phenotype correlation. Page 8-Lines 290-294

 Minor points:

  • Functional test: in terms of genetics and genomics, the hormonal experiments conducted by the authors do not constitute functional experiments. Since the authors carried out no study in animal or cell models, or even gene expression experiments, it is erroneous to use the term “functional” in the title, result, discussion, and elsewhere in the manuscript in relation to the effect of the GHS-Rand other variants observed.

ANSWER: Thanks for mention it, this is certainly a mistake. We have change “functional” for the more appropriate term “dynamic test” throughout all the text.

  • Page 7 lines 274 to 275: this statement is problematic: “due to the implications of GHS-R mutation in the stimulation tests”. How can a benign variant be responsible for these clinical observations?

ANSWER:  Taking into account that macimorelin is a ghrelin mimetic which  binds the GHSR receptor in the pituitary with an affinity similar to ghrelin, the flat GH response in the index case (homozygous) and a blunted peak of GH in the siblings (heterozygous) argue in favor that this variant inhibits or ameliorates the GHSR stimulatory effects of macimorelin on GH release throughout GHSR.  We have reinforced our statement by referring the studies by Pantel et al 2006, studies in mice and in the general population Page 7 lines 251-271

We have changed the term “stimulation” for “dynamic test”

  • The authors may want to use the term “variants” instead of “mutation” in the entire manuscript. The current consensus is that mutation should be used to describe the process that generates the genetic variants.

ANSWER:  Thank you for this comment. We have corrected all the text accordingly.

 Specific comments:

Page 1 line 18: remove the comma after “both”. Done

 Page 2 line 58: replace “high” with “height”. Done

 Page 2 line 65: replace “high” in the figure legend with “height”. Done Thanks!

 Page 3 line 110:  Insert “it” before the “was”. Done

 Page 3 line 122: delete the comma after “nutritional”. Done

 Page 3 line 123: delete the comma after “revealed”. Done

Page 3 line 125: you may want to insert a semi-colon after the “SDS”.

We fixed it (-2.85 SDS)

Page 3 lines 126 to 134: punctuations in these lines are poorly done.  

We have change this appropriately.

 Page 5 line 212: “Figure 3. Material)” may be presented as (Figure 3).

Done!

Page 8 line 303: the authors state that “Institutional Review Board Statement: Not applicable”. I sis the case that this research involving human subjects was carried out without IRB approval?

We have replaced the statement by the following:  The study was conducted according to the mandates of the Declaration of Helsinki and good clinical practices. The patients’ confidential information was protected according to the Spanish national data protection law.

Page 8 line 306: delete “Conflict of Interest”.

Done!

Page 8 lines 322 to 325: change the capital letters to sentence cases to reflect the formatting style for the references.

Done!

Reviewer 2 Report

This is an interesting manuscript. This is the first report of a patient with two extremely 265 rare mutations in a homozygous state ADAMTS17 and GHS-R. Few requirements are needed

1-English editing

2-Please  write the abbreviations in full then the abbreviation between brackets

3-Please add new references during 2023

English editing is needed

Author Response

#Reviewer 2#

This is an interesting manuscript. This is the first report of a patient with two extremely rare mutations in a homozygous state ADAMTS17 and GHS-R. Few requirements are needed.

ANSWER: Many thanks for your kind comments on our manuscript

1-English editing: Done

2-Please  write the abbreviations in full then the abbreviation between brackets.  Done

3-Please add new references during 2023. We have added this new relevant reference marked in yellow

Hassani, M.; Taghizadeh, S.; Farahzad Broujeni, A.; Habibi, M.; Banitalebi, S.; Kasiri, M.; Sadeghi, A.; Nozari, A. A Novel Missense Mutation in the TGF-β-Binding Protein-Like Domain 3 of FBN1 Causes Weill-Marchesani Syndrome with Intellectual Disability. Adv Biomed Res 2023, 12, 114, doi:10.4103/abr.abr_138_22.

Reviewer 3 Report

The manuscript entitled” Contribution of functional and genetic tests for short stature diagnosing: a case report”, a clinical case study focuses on an adult patient with short stature and explores the clinical phenotype and genetic characteristics associated with their condition. The patient exhibits severe short stature, small hands and feet, and eye disturbances. Genetic testing reveals homozygous variants in two genes: ADAMTS17, which is responsible for Weill-Marchesani-like syndrome, and GHS-R, which is involved in growth hormone (GH) regulation. Functional stimulation tests are conducted to assess the functional implications of the GHS-R mutation. An insulin tolerance test shows a normal GH response, while the GH response to macimorelin stimulation is found to be totally flattened. This highlights the functional involvement of the GHS-R mutation in the patient's short stature phenotype. The study emphasizes the importance of genetic factors in growth-related disorders and the challenges in establishing the clinical significance of genetic test results. It provides valuable insights into the molecular mechanisms and genetic basis of growth failure, expanding the understanding of rare genetic mutations. Additionally, the study discusses the implications of the identified mutations for genetic counseling, as they not only impact the index case but also have implications for her relatives. The identification of these mutations enables personalized treatment approaches and underscores the need for ongoing follow-up and management of individuals with growth failure. Overall, the authors are contributing to scientific knowledge by shedding light on the clinical phenotype, genetic characteristics, and functional implications of rare mutations in ADAMTS17 and GHS-R. It highlights the potential for genetic testing to improve diagnosis, management, and counseling for individuals with growth-related disorders. Further research in this area can help deepen our understanding and develop targeted treatments for these conditions.

While manuscript sheds light on important aspects of growth-related mutations, we have a few questions that could further enhance our understanding:

Limitations of the study

1. Small sample size: The study is based on a single clinical case, limiting the generalizability of the findings. The rarity of the identified mutations further restricts the ability to draw broad conclusions.

2. Lack of functional validation: While the study identifies the homozygous mutations in ADAMTS17 and GHS-R, there is a lack of detailed functional validation of these mutations. Further experiments or functional assays would strengthen the understanding of the molecular mechanisms and clinical implications.

3. Uncertain clinical significance of GHS-R variant: The clinical significance of the homozygous GHS-R variant (p.(Leu90Pro)) remains unknown. Although it was associated with a flat GH response in the index case and impaired macimorelin response in the siblings, further studies are needed to establish its functional implications and impact on growth.

4. Limited understanding of genotype-phenotype correlation: While the study describes the clinical phenotype associated with ADAMTS17 and GHS-R mutations, the precise relationship between the identified genetic variants and the observed phenotype is not fully elucidated. More comprehensive investigations are necessary to establish a clear genotype-phenotype correlation.

5. Absence of long-term follow-up: The study primarily focuses on the genetic and clinical characterization of the index case and provides limited information on the long-term outcomes, treatment efficacy, and potential complications associated with the identified mutations. Long-term follow-up studies are needed to assess the stability of the observed phenotype and the response to interventions.

6. Lack of control group: The absence of a control group hinders the ability to compare the identified mutations with a reference population. A control group would provide a better understanding of the specific contribution of these mutations to the observed phenotype.

7. Limited functional testing of GHS-R mutation: The functional testing conducted in the study, specifically macimorelin stimulation, provides insights into the GH response in the index case and siblings. However, additional functional tests could provide a more comprehensive understanding of the impact of the GHS-R mutation on growth regulation.

8. Lack of exploration of other genetic and environmental factors: The study focuses primarily on the ADAMTS17 and GHS-R mutations but does not extensively explore other potential genetic or environmental factors contributing to the observed phenotype. Considering the complexity of growth regulation, additional factors may play a role.

9. Potential confounding factors: The study does not account for potential confounding factors that could influence the observed phenotype, such as other genetic variants or environmental influences. Accounting for these factors would provide a more comprehensive understanding of the genetic contributions to growth failure.

10. Limited implications for clinical management: While the study provides valuable insights into the genetic characteristics of the index case and her relatives, the direct implications for clinical management, treatment options, and genetic counseling are not extensively discussed. Further research is needed to establish concrete recommendations for patient management based on these genetic findings.

Significant comments

1. What is the functional significance of the homozygous variant in GHS-R that was detected in the index patient?

2. Can you provide further insight into the molecular mechanism by which the ADAMTS17 mutation contributes to Weill-Marchesani-like syndrome and its association with short stature?

3. How do the homozygous and heterozygous mutations in GHS-R affect GH secretion and growth in the index patient and their siblings, respectively?

4. Have similar cases with homozygous mutations in both ADAMTS17 and GHS-R been reported previously, and if so, what were the clinical manifestations observed in those cases?

5. What are the implications of the flat GH response observed in the index patient during macimorelin stimulation, considering the involvement of GHS-R in GH regulation?

6. How do the identified mutations in ADAMTS17 and GHS-R contribute to the overall phenotypic spectrum observed in the index patient, including the presence of eye disturbances and other associated co-morbidities?

7. Have there been any studies exploring potential treatment options or interventions targeting the specific genetic mutations identified in this study?

8. How do the identified genetic mutations in ADAMTS17 and GHS-R interact with other genetic and environmental factors that influence growth, such as fetal growth restriction and nutrition?

9. Can the findings from this study be generalized to a broader population, or are they specific to this particular patient and their familial context?

10. Are there any potential genetic modifiers or additional genetic variants that may contribute to the observed phenotype in the index patient, and if so, how might they interact with the ADAMTS17 and GHS-R mutations?

11. How can genetic diagnosis be useful for better follow-up of siblings or offspring with potential consequences on growth due to inherited defects like those found in this case study?

Need to rectify

Author Response

#Reviewer 3#

The manuscript entitled” Contribution of functional and genetic tests for short stature diagnosing: a case report”, a clinical case study focuses on an adult patient with short stature and explores the clinical phenotype and genetic characteristics associated with their condition. The patient exhibits severe short stature, small hands and feet, and eye disturbances. Genetic testing reveals homozygous variants in two genes: ADAMTS17, which is responsible for Weill-Marchesani-like syndrome, and GHS-R, which is involved in growth hormone (GH) regulation. Functional stimulation tests are conducted to assess the functional implications of the GHS-R mutation. An insulin tolerance test shows a normal GH response, while the GH response to macimorelin stimulation is found to be totally flattened. This highlights the functional involvement of the GHS-R mutation in the patient's short stature phenotype. The study emphasizes the importance of genetic factors in growth-related disorders and the challenges in establishing the clinical significance of genetic test results. It provides valuable insights into the molecular mechanisms and genetic basis of growth failure, expanding the understanding of rare genetic mutations. Additionally, the study discusses the implications of the identified mutations for genetic counseling, as they not only impact the index case but also have implications for her relatives. The identification of these mutations enables personalized treatment approaches and underscores the need for ongoing follow-up and management of individuals with growth failure. Overall, the authors are contributing to scientific knowledge by shedding light on the clinical phenotype, genetic characteristics, and functional implications of rare mutations in ADAMTS17 and GHS-R. It highlights the potential for genetic testing to improve diagnosis, management, and counseling for individuals with growth-related disorders. Further research in this area can help deepen our understanding and develop targeted treatments for these conditions.

ANSWER: Many thanks for the revision process of our manuscript and your kind comments. We have addressed all your comments (see below), which has significantly improved the quality of our paper.

While manuscript sheds light on important aspects of growth-related mutations, we have a few questions that could further enhance our understanding:

Limitations of the study

  1. Small sample size: The study is based on a single clinical case, limiting the generalizability of the findings. The rarity of the identified mutations further restricts the ability to draw broad conclusions.

ANSWER: We agree with the reviewer’s concern that a larger sample would give stronger support to our results and, therefore, we have added this  as a limiting factor of our study. pag 8  line 309-314

  1. Lack of functional validation: While the study identifies the homozygous mutations in ADAMTS17 and GHS-R, there is a lack of detailed functional validation of these mutations. Further experiments or functional assays would strengthen the understanding of the molecular mechanisms and clinical implications.

ANSWER:  We agree with this comment understand the referee concern and we have remarked this issue in the revised manuscript in pag 7 lines 278-283 pag 8  line 309-314.

  1. Uncertain clinical significance of GHS-R variant: The clinical significance of the homozygous GHS-R variant (p.(Leu90Pro)) remains unknown. Although it was associated with a flat GH response in the index case and impaired macimorelin response in the siblings, further studies are needed to establish its functional implications and impact on growth.

ANSWER:  We agree with the referee and have added this statement in page 7 lines 278-283

  1. Limited understanding of genotype-phenotype correlation: While the study describes the clinical phenotype associated with ADAMTS17 and GHS-R mutations, the precise relationship between the identified genetic variants and the observed phenotype is not fully elucidated. More comprehensive investigations are necessary to establish a clear genotype-phenotype correlation.

ANSWER: We agree with the referee that this point merit a better explanation. We have added this paragraph to the revised manuscript: “Therefore, our patient has two defects that impact on stature, the alteration of the connective tissue (i.g. growth plate) from ADMTS17 variant and the impaired GH secretion from GHSR variant, both variants could contribute to the extreme SS phenotype observed in the index case. However, more comprehensive investigations to establish a clear genotype-phenotype correlation are needed”. Pag 8 Lines 290-294

  1. Absence of long-term follow-up: The study primarily focuses on the genetic and clinical characterization of the index case and provides limited information on the long-term outcomes, treatment efficacy, and potential complications associated with the identified mutations. Long-term follow-up studies are needed to assess the stability of the observed phenotype and the response to interventions.

ANSWER: In the index case which has the growth plate close there is not possible medical  intervention for improving height. However, these two main lessons can be drawn of this case:

  1. Because of potential consequences on younger relatives and offspring is relevant to perform genetic test in  adult patients with short stature and familial  history of short stature and or syndromic features, 
  2. To identify GHS-R mutations has clinical relevance, because those patients with this mutation have unaffected response to standards dynamics tests (e.g. insulin tolerance test) which work by GHRH-R pathways. Thus, a patient with GHSR variant could be misdiagnosed and will not obtain any  benefit from GH treatment

We have added these two points to the clinical case in page 8 lines 301-307 of the revised manuscript

  1. Lack of control group: The absence of a control group hinders the ability to compare the identified mutations with a reference population. A control group would provide a better understanding of the specific contribution of these mutations to the observed phenotype.

Answer: The nature of our manuscript (case-report) precludes to have a control. However, this is a limiting factor that has been added to the discussion of the revised manuscript.  Page 8 lines 308-313

  1. Limited functional testing of GHS-R mutation: The functional testing conducted in the study, specifically macimorelin stimulation, provides insights into the GH response in the index case and siblings. However, additional functional tests could provide a more comprehensive understanding of the impact of the GHS-R mutation on growth regulation.

ANSWER: We agree with the referee and we have stated that further studies are needed to establish the functional implications and impact on growth of GHSR page 8 lines 278-284 and lines 291-295 and 308-3013

  1. Lack of exploration of other genetic and environmental factors: The study focuses primarily on the ADAMTS17 and GHS-R mutations but does not extensively explore other potential genetic or environmental factors contributing to the observed phenotype. Considering the complexity of growth regulation, additional factors may play a role.

ANSWER: In this regard, first we performed a targeted approach for some candidates genes related with hight  ACAN, BRAF, BTK, CBL, CCDC8, COL10A1, COL11A1, COL11A2, COL1A1, COL2A1, COL9A1, COL9A2, COL9A3, CUL7, EVC, FBN1, FGFR3, GH1, GHR, GHSR, GHRHR, GLI2, GPC3, H19, HESX1, HRAS, IGF1, IGF2, IGF1R, IGFALS, IHH, KRAS, LHX3, LHX4, MAP2K1, MAP2K2, MRAS, NF1, NPPC, NRAS, OBSL1, OTX2, POU1F1, PPP1CB, PROKR2, PROP1, PTPN11, RAF1, RIT1, RRAS, SHOC2, SHOX, SOS1, SOS2, SOX3, SPRED1, SRCAP, STAT5B -> here we identify the GHSR variant. Then we followed with the Human Phenotype Ontology terms HP:0004322, HP:0032367, HP:0011003, HP:0004209, HP:0001156 -> here we identify the ADMATS17 variant. Finally in the exome extension (non-target study) we did not identify any other variant than ADAMTS17 or GHS-R that could be linked with the phenotype. We have amended the text page 4 line 151-152 and and pag 8 lines 294-295

  1. Potential confounding factors: The study does not account for potential confounding factors that could influence the observed phenotype, such as other genetic variants or environmental influences. Accounting for these factors would provide a more comprehensive understanding of the genetic contributions to growth failure.

ANSWER:  We have commented on this issue in the revised manuscript: e page 4 line 151-152 and pag 8 lines 294-295 and 308-313

  1. Limited implications for clinical management: While the study provides valuable insights into the genetic characteristics of the index case and her relatives, the direct implications for clinical management, treatment options, and genetic counseling are not extensively discussed. Further research is needed to establish concrete recommendations for patient management based on these genetic findings.

ANSWER:  Thanks for mention it. We have added the following recommendations regarding management and genetic counseling to the revised manuscript

1.- Direct implications in management and treatment: The diagnosis of both variants was helpful. For the patient it offered her better follow-up related to ADAMTS17 such as supportive treatment for eye disturbances and other co-morbidities.  For her siblings the genetic study ruled out Weill-Marchesani like syndrome (they were heterozygous) and uncovered a GHS-R variant which is important to know in order to initiate GH treatment if required and avoid the misdiagnosis of a GHD. We have added this comment on Pag. 8  295-300 of the revised manuscrip

2.-The genetic counseling, lies on the extreme rare and recessive form  of both variants, leading to an homozygous variant only in small communities with co-sanguine events. However, further research is need to give specifics recommendations Added on Page 8 line 308-313of the revised manuscrip

Significant comments

  1. What is the functional significance of the homozygous variant in GHS-R that was detected in the index patient?

Answer: A potential physiological role of ghrelin in the regulation of GH release is supported by the data of Pantel et al. (2006)  doi:10.1172/JCI25303 showing that  a mis-sense mutation which impairs the constitutive activity of the GHSR was  associated with short stature. (page 7 lines 251-262)

Similarly, GHS-R-null mice have lower IGF-I levels when compared to wild type animals  and ghrelin null mice have a tendency to lower IGF-I concentrations when compared to the control animals (Sun et al., 2003). Moreover, the application of a GHS-R antagonist results in a decrease in GH pulse amplitude in rodents Zizzari et al. (2005). We have added this new information to the revised manuscript ( page 7 lines 263-266).

In the general population macimorelin binds to GHS-R and stimulates dose-dependent increases in GH levels, which are used to assess GH production in patients with suspected GH deficiency. This response was affected in this family report Page 7 lines 267-272

Finally, the functional significance lies on that patients with this homozygous variant have impaired GH secretion which cannot be identified by using  the standard dynamic tests (ITT or DOPA) Thus, the patients with GHSR variants could be misdiagnosed or overlooked ( Pag 8 lines 278-284 and 304-307 of the revised manuscript).

  1. Can you provide further insight into the molecular mechanism by which the ADAMTS17 mutation contributes to Weill-Marchesani-like syndrome and its association with short stature?

Answer: Thanks for request i. We have given further insight on the molecular mechanism of ADAMTS17 in page 5 lines 205-213

  1. How do the homozygous and heterozygous mutations in GHS-R affect GH secretion and growth in the index patient and their siblings, respectively?

Answer: Taking into account that macimorelin is a ghrelin mimetic which  binds the GHSR receptor in the pituitary, the flat GH response in the index case (homozygous) and a blunted peak of GH in the siblings (heterozygous) argue in favor that this variant inhibits or ameliorates  the GHSR  stimulatory effects of macimorelin  on GH release throughout GHSR.  page 8 Lines 278-284

  1. Have similar cases with homozygous mutations in both ADAMTS17 and GHS-R been reported previously, and if so, what were the clinical manifestations observed in those cases?

Answer: To the best of our knowledge this is the first report of a patient with two extremely rare mutations in a homozygous state ADAMTS17 and GHS-R . Page 8 Lines 315-316

  1. What are the implications of the flat GH response observed in the index patient during macimorelin stimulation, considering the involvement of GHS-R in GH regulation?

Answer: The flat response in the index case has 2 main clinical implications (1) patients with this homozygous mutations will have impaired GH secretion (ghrelin increase GH release). (2) The patients could be misdiagnosed because  the standard functional tests ( ITT or DOPA)  works via  GHRH-R pathways, which are  unaffected. We have address this issue in the revised manuscript ( page 8 Lines 284-289).

  1. How do the identified mutations in ADAMTS17 and GHS-R contribute to the overall phenotypic spectrum observed in the index patient, including the presence of eye disturbances and other associated co-morbidities?

Answer: ADAMTS17  is related with Weill-Marchesani Syndrome (WMS)  a rare connective-tissue disorder characterized of short stature, eyes disturbances consisting of lens subluxation, spherophakia, severe myopia and possible glaucoma secondary to shallow anterior chamber angles, ADAMTS17  is also related to incomplete forms of WMS as in our patient ( called WMS-Like). On the other hand, GHS-R mutation and consequently impaired GH release could be behind the extreme short stature of our patient. Therefore, our patient has two defects that impact in stature, alteration of the connective tissue (i.g. growth plate) from ADMTS17 variant and impaired GH secretion from GHS-R page 8 Lines 291-295

  1. Have there been any studies exploring potential treatment options or interventions targeting the specific genetic mutations identified in this study?

Answer: To the best of our knowledge there is not specific treatment for ADAMTS17 mutation, but  supportive treatment of  eye disturbance and other comorbidities could have a better approach knowing the diagnostic. On the other hand, patients with GHSR  mutation could improve their growth velocity after GH treatment (Pantel 2006) doi:10.1172/JCI25303.

  1. How do the identified genetic mutations in ADAMTS17 and GHS-R interact with other genetic and environmental factors that influence growth, such as fetal growth restriction and nutrition?

Answer: To the best of our knowledge there are no reports regarding this mutations and  fetal growth restriction.

  1. Can the findings from this study be generalized to a broader population, or are they specific to this particular patient and their familial context?

Answer: Our results have to be taken with caution due to the extremely rare of the co-existence in a single patient of these two rare variants. Nevertheless,  given that both variants affect very different mechanisms and both related to growth, it is very likely that the effects were independent and synergistic.

  1. Are there any potential genetic modifiers or additional genetic variants that may contribute to the observed phenotype in the index patient, and if so, how might they interact with the ADAMTS17 and GHS-R mutations?

 Answer: We did not find  other genetic variant in the index case, thus we think that this possibility is very unlikely.

  1. How can genetic diagnosis be useful for better follow-up of siblings or offspring with potential consequences on growth due to inherited defects like those found in this case study?

 Answer: The diagnosis of both variants was helpful. For the patient it offered her better follow-up related to ADAMTS17 as supportive treatment for eye disturbances and other co-morbidities.  For her siblings the genetic study ruled out Weill-Marchesani like syndrome (were heterozygous) and uncovered GHS-R variant which is important to know in order to  initiate GH treatment if required and avoid the misdiagnosis of a GHD. Page 8 lines 296-300

Round 2

Reviewer 1 Report

Introductory paragraph: The authors have largely addressed the comments I made in my previous submissions. Where they could not address the comments experimentally, they responded to such queries as limitations to their study. Unfortunately, the authors are still silent on IRB approval for the study. Please see below my general and specific comments.

General comments:

Major points:

·       Page 2 lines 73 to 74: The authors are STILL silent on Institutional Review Board (IRB) approval for the study, though they present information on consent. The authors may want to add a statement stating that IRB approval was obtained (if indeed that is the case) and include the approval number and the name of the IRB that gave the approval. It will not be acceptable to carry out such human subject research without IRB approval.

Minor points:

·       None

Specific comments:

Page 1 line 16: replace “mutations” with “variants”.

Page 2 line 65: replace “heigh” in the figure legend with “height”.

Page 7 line 251: replace “mutation” with “variant”.

Page 7 line 281: consider replacing “as the responsible of this” to “may be responsible for”.

Page 9 lines 358 to 360: The authors still could not state emphatically that the study was reviewed and approved by an Institutional Review Board (IRB). Every research study must be reviewed and approved by an IRB and the statements they provide on this are still vague. They must state the name of the IRB that gave the approval, as well as the IRB approval number.

Page 9 line 363: delete “Conflict of Interest”.

Page 9 line 366: do not delete “Conflicts of Interest: The authors declare no conflict of interest.”

Author Response

Introductory paragraph: The authors have largely addressed the comments I made in my previous submissions. Where they could not address the comments experimentally, they responded to such queries as limitations to their study. Unfortunately, the authors are still silent on IRB approval for the study. Please see below my general and specific comments.

Answer: Many thanks for the revision process. We have highlighted in yellow the new requirement  

General comments:

Major points:

  • Page 2 lines 73 to 74: The authors are STILL silent on Institutional Review Board (IRB) approval for the study, though they present information on consent. The authors may want to add a statement stating that IRB approval was obtained (if indeed that is the case) and include the approval number and the name of the IRB that gave the approval. It will not be acceptable to carry out such human subject research without IRB approval.

Answer: This is a clinical case that follows our standard clinical practice without any diagnostic test or treatment out of our usual protocols.  In accomplishment to the regulations of our Hospital the IRB approval is not required.  In fact, our Ethics Committee prevents to evaluate this type of single cases and only recommend to obtain the written consent from the patient when a publication is planned.  We have added a sentence to the revised manuscript clarifying this issue page 2 Lines 71-74

Minor points:

  • None

 Specific comments:

Page 1 line 16: replace “mutations” with “variants”. Done!

Page 2 line 65: replace “heigh” in the figure legend with “height”. Done. Thanks!

Page 7 line 251: replace “mutation” with “variant”. Done!

Page 7 line 281: consider replacing “as the responsible of this” to “may be responsible for”. We made the replacement as suggested.

Page 9 lines 358 to 360: The authors still could not state emphatically that the study was reviewed and approved by an Institutional Review Board (IRB). Every research study must be reviewed and approved by an IRB and the statements they provide on this are still vague. They must state the name of the IRB that gave the approval, as well as the IRB approval number.

Answer: We have explained above. We have changed the Institutional Review Board Statement declaration: as follows:  This clinical report followed our usual clinical practice and, therefore, the IRB approval was not required.

Page 9 line 363: delete “Conflict of Interest”. Done!

Page 9 line 366: do not delete “Conflicts of Interest: The authors declare no conflict of interest.” Done!